# Improving best practice for patients receiving hospital discharge letters: a realist review protocol

Katharine Weetman,[1] Geoffrey Wong,[2] Emma Scott,[1] Stephanie Schnurr,[3] Jeremy Dale[1]

[1]Division of Health Sciences, Primary Care, Warwick Health Sciences, University of Warwick, Coventry, UK
[2]Nuffield Department of Primary Care Health Sciences, Medical Sciences Division, University of Oxford, Oxford, UK
[3]Centre for Applied Linguistics, University of Warwick, Coventry, UK

**Correspondence to**
Katharine Weetman;
K.Weetman@warwick.ac.uk

## ABSTRACT

**Introduction** Discharge documents are important for transferring information from hospitals to the referring clinician; in the UK and many countries, this is often the patient's general practitioner or family physician. However, patients may or may not receive their discharge letters, and whether patients should routinely receive discharge letters remains unclear.

**Methods and analysis** The review will consolidate evidence on patients receiving discharge letters through the theory-driven approach of a realist review. The review will be conducted systematically and seek to explain how, why, for whom and in what contexts does this practice 'work'. The review will specifically explore whether there *are* benefits of this practice and if so what are the important contexts for triggering the mechanisms associated with these outcome benefits. Negative effects will also be considered. Several steps will occur: devising initial rough programme theory, searching the evidence, selecting relevant documents, extracting data, synthesising and finally programme theory refinement. As the process is viewed as iterative, this cycle of steps may be repeated as many times as is necessary to reach *theoretical saturation* and may not be linear. The initial programme theory will be tested and refined throughout the review process and by stakeholder involvement of National Health Service (NHS) policy makers, practitioners and service users.

**Ethics and dissemination** Formal ethical review is not required. The resulting programme theory is anticipated to explain how the intervention of patients receiving written discharge communication may work in practice, for whom and in what contexts; this will inform best practice of patients receiving discharge communication. The review findings will be disseminated in a peer-reviewed journal and presentations and discussions with relevant organisations and stakeholders. While the review will be from the perspective of the UK NHS, its findings should be relevant to other healthcare systems.

**PROSPERO registration number** CRD42017069863.

## Strengths and limitations of this study

► This is the first realist review to synthesise and develop theories about patients receiving written discharge communication.
► A realist review approach accounts for complexity, which is relevant and apt for research relating to improving healthcare policy, a complex process.
► The engagement of patients, general practitioners and policy makers in refining the programme theory will ensure its relevance to different stakeholder perspectives.
► Given limited study resources, wider stakeholder involvement is not feasible.
► Only studies published in the English language will be included.

physician.[1] This is particularly important in healthcare systems in which primary care services are well established, such as in the UK. 'Discharge communication' may follow inpatient or outpatient discharge and typically comprises a discharge letter or summary. Sometimes the patient may also receive written discharge communication, but in the UK this is not standardised.

In 2003, the Department of Health (DH) in England released 'good practice guidelines' recommending that National Health Service (NHS) patients should be copied into their letters where appropriate.[2] This was intended to increase patient understanding, the quality of information sent and improve doctor–patient relationships.[1–4] However, the evidence of how patients feel about this and moreover whether this practice *is* beneficial and, if so, *when, how* and *for whom* remains limited. Evidence from the UK and other settings indicates that patients receiving medical letters can be beneficial,[4–15] with outcomes including: increased understanding,[6] increased patient satisfaction,[8] reduced readmissions[15] and increased patient involvement in their care.[9] There is also high reported preference by patients for receiving letters

## INTRODUCTION
### Background

It is a well-established practice that written *discharge communication* should take place between the discharging physician and follow-up physician, typically the patient's general practitioner (GP) or family

(94% where n=63,[6] 95% where n=500).[5] However, there are UK studies[7 10 16 17] and non-UK studies[18 19] that suggest 'detriments' or concerns with patients receiving letters, including: concerns over confidentiality,[7 17] potential patient distress with letter content,[7] associated financial costs to the NHS,[7 17] issues around the comprehensibility of medical letters[16 17 19] and failing to acknowledge the voice of patients who do not want to receive letters.[7]

Recently, there have been studies, both within[20] and outside the UK,[13 18] that move beyond simply 'copying' patients into correspondence and instead writing 'patient-directed letters'. In 2014, Bench *et al*[20] explored the feasibility and effects of giving patients personalised discharge summaries produced by nurses. They found the summaries helped support patients and increased patient understanding. Nonetheless, barriers were identified for implementing this intervention such as 'motivation' and 'time constraints'.[20] Similarly, in 2016, Buurman *et al*[18] looked at personalised patient discharge letters. Although the practice was generally rated 'positively' by patients and physicians in their research, they reported medical interns felt 'explaining medical terms in understandable plain language was a difficult task' and one which incurred a feeling of 'great responsibility' and insecurity.[18]

In summary, whether it *is* beneficial for patients to receive written discharge communication, and, if so, *for whom, when, how, why* and whether this should be a direct copy or personalised letter remains equivocal. We could find no review specific to this question; we only found reviews of copying letters in general, for example, Minhas[8] and Harris and Boaden.[12] We therefore concluded that formal consolidation of the evidence is required.

## Realist review methodology

A realist review may be defined as a, 'theory-driven, interpretative approach to the synthesis of evidence'.[21] The evidence synthesised may be qualitative, quantitative or mixed methods.[22] In line with taking a theory-driven approach, one of the main steps of a realist review, as outlined in the work of Pawson,[23–25] is to develop and refine a 'middle-range' realist *programme theory* that details how an intervention or programme may be theorised to 'work' as well as under what contexts, for whom, why and to what extent. Thus, this review seeks to develop a 'programme theory' for patients receiving written discharge communication.

A realist review approach views 'causation' as *generative*, that is, 'mechanisms' may be triggered within certain 'contexts' resulting in one or more 'outcomes' following an event or 'intervention'.[24] A realist review, therefore, is valuable to inform attempts to reproduce beneficial or positive outcomes through understanding how an intervention *works* and hence under what circumstances, the mechanisms connected to beneficial outcomes may be triggered.[26] Hence, within a healthcare context, a realist review can aid understanding and explanation of how the intervention may improve clinical outcomes.

Another value or strength of a realist review is the capacity to account for *complexity* and non-linear causal relationships; this is particularly relevant for research on the intervention of patients receiving written discharge communication.[21 23] The intervention under scrutiny is complex in several ways: the *form* of discharge communication may vary, and the success of the intervention is highly context dependent and most likely influenced by factors such as practitioner communicative competence, patient education and understanding, and attitudes of the patient and professional.

A realist review aims to *explain* how and why an intervention may be theorised to *work* (or not).[24] The notion of moving beyond effectiveness *evaluation* of an intervention and onto *explanation* of how and why an intervention *works* is one of the key distinctions between a realist review and other traditional review types such as a systematic review; it is also one of the realist review strengths in application to healthcare and social policy.[23 27] Due to the well-documented strengths of realist reviews, it is perhaps unsurprising that realist reviews are being increasingly used within healthcare contexts (eg, refs[21 22 28–34]). Thus, a realist approach is suitable and useful for the current research.

## METHODS AND ANALYSIS
### Review aim, questions and objectives
#### Aim

This study aims to understand how and why the different effects are produced from patients receiving written discharge communication.

Effects may be simplified into desired/intended or 'positive' and undesired or 'negative' depending on whether the outcome is reported in the source as beneficial (eg, increased patient understanding of condition) or detrimental (eg, increased patient anxiety).

#### Research questions (RQs)

RQ1: what positive and negative effects have been reported on patients receiving written discharge communication?

RQ2: what are the important contexts that determine whether the different mechanisms produce positive and negative effects, and why?

#### Objectives

1. To conduct a realist review to understand how and why the different effects arise when patients receive written discharge communications.
2. Develop a programme theory for patients receiving written discharge communication.
3. To make recommendations for best practice for patients receiving written hospital discharge communication.

Review start date: June 2017.

Review anticipated completion: January 2018.

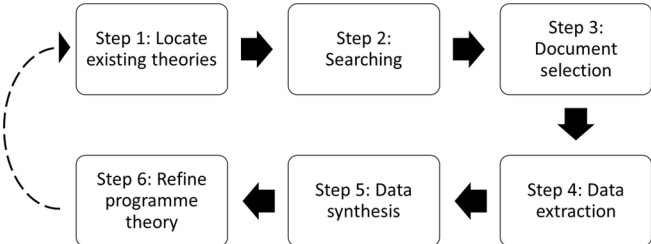

**Figure 1** Review design.

## STUDY DESIGN

The review design is based on a collation of Pawson's five review stages,[24] the project protocol by Ford *et al*[27] and the project diagram by Wong *et al*.[22] The design is summarised in figure 1.

### Step 1: locate existing theories

Locating existing theories on how patients receiving written discharge communication is theorised to work or not work in different contexts will be completed through a scoping search by KW. The scoping search will be based on search terms centred on the intervention under study (eg, patient copies/receiving letters/ discharge communication). This search will include a mix of electronic published resources (MEDLINE and Web of Science) and UK healthcare websites (DH and Royal College of Physicians). Documents sourced within the scoping search will be interrogated for theories relating to patients receiving discharge communication. Theories located in the scoping stage will be inspected and selected based on relevance to the review aims and RQs; we seek theories that aid *explanation* of how and why patients receiving discharge communication results in different positive effects (eg, drug adherence) and negative effects (eg, preventable hospital readmissions).

Any initial scoping search done to build the initial programme theory is not meant to be exhaustive but to function as a starting point for the realist review. During the review, the initial programme theory is gradually developed. There are no hard and fast rules for how well developed the initial programme theory needs to be before the main searching is undertaken. Instead, judgement is needed as is the need to balance the degree of comprehensiveness and practicalities. Our decision is that due to feasibility limitations, no more than 30 documents will be screened for theories in step 1.

Any search strategies detailed in documents found from the scoping search will be used to inform step 2. 'Keywords', 'Medical Subject Headings (MeSH)' and any other indexing for the documents found will also be used to inform the step 2 searching phase. From the findings of the scoping search and using the various content expertise of the research team, an initial 'rough' programme theory will be developed, to be refined throughout the realist review process. Once the initial programme theory has been developed, step 2, the structured formal searching phase, will commence.

**Table 1** Search sources.

| | Sources to be searched |
|---|---|
| 1 | MEDLINE |
| 2 | EMBASE |
| 3 | CINAHL |
| 4 | DARE |
| 5 | ASSIA |
| 6 | Web of Science |
| 7 | ZETOC |
| 8 | AMED |
| 9 | NHS Digital (HSCIC) |
| 10 | NHS Evidence (public domain only) |
| 11 | DH |
| 12 | NICE Guidelines |
| 13 | Cochrane Database of Systematic Reviews |
| 14 | EPPI-CENTRE |
| 15 | SCOPUS |
| 16 | Google Scholar |
| 17 | OpenGrey |
| 18 | Greynet sources |
| 19 | ProQuest dissertations and theses |
| 20 | General Medical Council |
| 21 | Royal College of Physicians |
| 22 | Local Medical Committees (West Midlands) |
| 23 | Clinical Commissioning Groups (West Midlands) |
| 24 | SIGN |

### Step 2: searching

Following Sholl *et al*,[34] the list of search terms will be first piloted and modified using Medline by KW and an information specialist. Thereafter, the modified list of search terms will be employed and adapted as required across source types. The searching phase will entail a 'purposive' sampling strategy using an iterative approach.[23] In line with a realist approach, the search strategy is intended to include a diverse range of evidence for programme theory development and refinement. The search will be information specialist led. Search terms will be guided by 'keywords', 'MeSH terms', topic indexing and any found search strategies from documents located in step 1. We anticipate the search strategy will need further testing and modification during the searching phase.

Electronic and manual searching will take place. Material included will be sourced from electronic databases, UK healthcare sites, grey literature searching and publications and archives of collaborating and local commissioners and policy makers (see table 1). In addition, hand-searching of bibliographies, 'cited by' searching and contacting experts will also be undertaken.

The search strategy is not intended to be fully comprehensive or exhaustive, but should provide a large enough

overview of the literature and sources to be meaningful and develop and refine the programme theory.[23]

### Step 3: document selection

Documents will primarily be selected on their *relevance*; they must contain data that inform the programme theory.[23 24 35] Crucially, as explained by Brennan *et al*,[21] this does not mean the entirety of the document must inform the programme theory but that the selection process will 'consider small sections of the primary study to test a very specific hypothesis about the relationships between context, mechanism and outcomes'. Assessing the relevance of documents in the selection phase will be discussed and decided among the research team. Hence, selection of documents will be grounded in whether they provide knowledge to the theory of how patients receiving discharge communication works.

Reference manager software will be used to export citations of search results. Search results will be screened and selected first by title, second by abstract and finally by the full text. Data screening and selection will be undertaken by KW with another member of the review team, who will assume the role of second reviewer. KW will screen the full set of search results. The second reviewer will screen a selection of 10% to check for consistency; we have chosen this proportion following the review by Wong *et al*.[22] Reasons for all exclusions will be recorded by all reviewers. A kappa measure will be calculated, and inter-reviewer disagreement (K<0.8) will result in the second reviewer screening the remaining respective 90%. Thereafter, the reviewers will discuss their selections until document inclusion consensus is reached for the phase. This process will take place for each of the screening and selection phases: titles and abstracts and full texts. The wider research team will adjudicate contested document selections if necessary.

The preliminary inclusion and exclusion criteria below will be applied by reviewers to all sources. As with all other steps in the realist review, screening and document selection will be viewed as an iterative process and so inclusion and exclusion criteria may change and develop.[28] The review intends to source quantitative, qualitative and mixed methods evidence. For the purposes of this review, the intervention 'patients receiving written discharge communication' will be defined as the patient being given or sent any form of written paper or hard copy hospital discharge communication or such communication being made available digitally; this may be a direct copy (cc:(PATIENT)), a patient-directed letter or a combination of the two.

Inclusion criteria:
► Meet 'relevance' criteria.[23 24]
► Patients discharged from hospital setting (inpatients and outpatients) to GP or family physician or community physician care.
► Discharge where written correspondence, 'discharge communication' is sent to GP OR GP and patient.
► Document/journal article/source written in English.

Exclusion criteria:
► Discharge communications to units or physicians other than GPs or family or community physicians for example, another hospital.
► Discharge of patients with conditions who lack cognitive capacity, for example, dementia, as their communicative needs are specialised.
► Discharge where no written communication took place, for example, telephone only.
► Patients <18 years.
► Discharge letters or summaries that are not in English or where the document details the patient required assistance reading their letter, for example, translation by a relative.

The preliminary criteria have been developed in order so that the resultant programme theory may encompass a variety of discharge types and be relevant across hospitals and specialities for a range of patients. However, the exclusion criteria inflicts limitations on the review. The first exclusion criterion states patient discharge communication to those other than GPs or family or community physicians will be excluded. This is because the review specifically focuses on discharge communication to GPs and patients rather than referrals or care handovers. It is only specifically discharge to units other than GPs/family physicians such as another hospital, for example hospital to hospital discharge, that is being excluded. Patients who may have particularly specialised communicative needs (eg, children) or where the intervention may have a higher risk of causing harm (eg, psychiatric discharge documents and dementia discharge documents) have been excluded; the communication needs of these patients may be more complex and variable within and between different patient groups and therefore is not possible within the review scope.

International evidence that meets the criteria will be considered. Consequently, data extraction and synthesis will carefully examine documents according to their geographical and healthcare system context.

The document selection process will be recorded with an adapted Preferred Reporting Items for Systematic Review and Meta-Analysis flow diagram to provide a clear audit trail.[36]

### Step 4: data extraction

Data extraction of the selected documents will be undertaken by KW, and the second reviewer. Realist review methodology primarily achieves data extraction through annotation and note-taking methods rather than finite or fixed data extraction forms.[21 23 34] Comparably with Mills *et al*[37] and Wiese *et al*,[38] we propose a hybrid approach to data extraction; characteristics of the documents will be recorded in a data extraction form in Excel and annotation of the full texts for programme theory ideas and subsequent labelling will be undertaken manually. The hybrid approach is useful in providing descriptive information for grouping documents during synthesis while still grounding data extraction in commonly used realist note-taking techniques.[23 24]

The data extraction form will record: basic information (authors, year of publication, ID or reference, source type and where and/or how the document was sourced), document context (geographical location and healthcare system details), document details (aims, design, methods, setting, findings and conclusion), intervention details (type of intervention or programme, eg, direct patient copy of discharge letter, number of participants, clinical specialty, participant details, form of discharge communication (eg, discharge summary) and who was involved in intervention process). We anticipate not all of the above details will be recordable for each document; we aim to record pertinent document characteristics. Data extraction forms will first be piloted and refined as needed. Completed data extraction forms will be discussed and checked with the research team for accuracy; adjustments to the form may be made.

The annotation phase and note-taking methods will be guided by the rough programme theory developed in step 1; we will test and refine the theory using data from included documents.[37] The two members of the research team will manually review, examine, highlight and annotate the documents in relation to context, mechanism and outcome (CMO) information and any theories about how the intervention does or does not work. In line with the work of Pawson et al,[23] documents will be 'scrutinised for which programme idea they address' and labelled. Annotations will be consolidated and discussed among the two annotators and wider research team.

During the data extraction phase, documents will also be quality appraised for Pawson's concept of *rigour*.[23] Brennan et al[21] describe *rigour* as 'whether the methods used to generate the relevant data are credible and trustworthy'. Rigour will be assessed in accordance with 'Realist and Meta-Review Evidence Synthesis: Evolving Standards' (RAMESES) guidelines and standards.[39 40] It is important to note that although we will assess 'rigour', Pawson et al[23] advise against exclusion of an entire document *solely* based on rigour; they argue this could, 'reduce rather than increase the validity and generalisability of review findings' as different parts of different documents contribute to the evidence base for programme theory testing and refinement. Hence, we will only make judgement about the rigour of data that we have assessed to be relevant for programme theory development and testing.[23]

### Step 5: data synthesis

Data and analysis from step 4 will be consolidated and synthesised to refine the programme theory by KW and the wider research team. A realist analytic approach will be used to interrogate the theory, according to Pawson et al,[24] and assess, according to the data, what 'works', why, for whom, to what extent and in what circumstances. Specifically, during data synthesis, we will look at evidence of the different outcomes within the initial programme theory and infer how these are caused in certain contexts through triggering different mechanisms.[21 23] We will be

using the framework for synthesising evidence termed by Pawson et al[23] as 'synthesis to consider the same theory in comparative settings'. They explain, 'this approach to synthesis assumes that particular programme theories work in some settings and not others and aims to make sense of the patterns of winners and losers'. We assume that the theory of patients receiving discharge communication does work in particular settings in particular forms but not in others and therefore that there may be different effects this intervention has depending on context. Hence, this approach is advantageous through comparing the intervention in the various settings found within the included documents. Our 'hybrid' approach to data extraction will aid the comparative process through permitting rapid 'groupings' of intervention settings alongside the programme theory annotations and labelling from step 4.

Relevant data from each document will be systematically considered to test and refine the programme theory using the following analytical strategies[21 38 39]:

► Juxtaposition of data sources: align sources and use evidence of each to build on and clarify each other.
► Reconciliation of data discrepancies: examine and explore reasons for apparent disparities between data.
► Adjudication of data: quality consideration on the foundation of methodological strengths and weaknesses.
► Consolidation of data: inference of mechanisms for outcomes.
► Situation of evidence: consideration of details of settings in order to complete 'context' element of CMOs and explain differing outcomes of intervention.

To address the research questions, we will cross-tabulate and compare the CMOs in order to highlight patterns of the important contexts for positive and negative effects and any reported benefits of the intervention. CMOs will be consolidated through the process of cross-tabulation and subsequently integrated into the programme theory.

The research team is made up of healthcare researchers, practising healthcare professionals, social scientists and medical students; this range of expertise is expected to facilitate and promote rigorous analysis and synthesis of data.

### Step 6: refine programme theory

The final stage is the refinement and testing of the programme theory in light of the synthesised data.[23] Stakeholder perspectives will assist refinement of the final theory through providing 'content expertise'.[22] Brennan et al[21] describe stakeholder contributions as a 'reality check' to test whether the programme theory derived from the published literature aligns with stakeholder experiences in practice. Stakeholders will primarily be engaged through the project collaborating Clinical Commissioning Groups (CCGs for South Warwickshire and Coventry & Rugby) and West Midlands Clinical Research Network. We will limit stakeholder involvement to three different groups: policy and decision makers,

practitioners and service users. These will be included due to their differing perspectives on the discharge communication process.

We aim to hold three discussion groups; one for each stakeholder type. KW will lead the discussion sessions with a research assistant. During these sessions, stakeholders will be able to view the results and analyses of the review with the opportunity to influence interpretation of findings and refine analyses; the sessions will particularly focus on discussing and refining the programme theory. It is pertinent to consult stakeholders to increase the relevance and practicability of the review recommendations for informing best practice. Formal ethical approval will not be required but informed participation will be sought.

After completion of step 6, any or all of the review steps may be revisited as necessary to refine the programme theory and attain 'theoretical saturation'. The threshold for 'theoretical saturation' will be decided according to Pawson's 'test of saturation'.[24 35] Consequently, after each cycle of review steps, the research team will determine whether the latest cycle has provided additional information about the intervention to answer the research questions and test the programme theory.[28] As such, the stopping point for the review will be determined when 'theoretical saturation' is reached; when the addition of documents and repetition of steps is not adding further knowledge.[24]

A diagram will be used to present the final programme theory alongside a narrative summary. The review will be reported according to RAMESES standards.[39 40]

## ETHICS AND DISSEMINATION
### Ethics
Formal ethical approval is not required for this review.

### Dissemination
The final theory will contribute towards explanation of what works in relation to patients receiving discharge communication. The findings will provide valuable insight into how and why this intervention produces its different effects that will support improved practice guidelines and policy. Informing best practice is of benefit to multiple stakeholders involved in sending and receiving discharge communication and development and regulation of this intervention. The review findings will be disseminated in a peer-reviewed journal, conference presentations and discussions with policy makers, educationalists and commissioners and relevant organisations to ensure the findings are readily available to inform best practice of patients receiving their hospital discharge letters. While the review will be undertaken from the perspective of UK NHS, its findings should be relevant to other healthcare systems in which there are well-developed primary care services.

**Acknowledgements** We would like to thank Samantha Johnson, our information specialist, for critically appraising the search strategy.

**Contributors** KW, ES, SS and JD conceptualised the study. KW is responsible for the design and drafting of the protocol manuscript. GW, ES, SS and JD contributed to protocol development and GW also provided methodological advice. GW, JD, ES and SS critically reviewed and edited the manuscript. All authors read and approved the final manuscript.

**Funding** This work is supported by the Economic and Social Research Council (ESRC) grant number ES/J500203/1 and Clinical Commissioning Groups (CCGs) of Coventry & Rugby and South Warwickshire. Funding for the open access charges for publication of this protocol was provided by the RCUK fund.

**Competing interests** None declared.

**Provenance and peer review** Not commissioned; externally peer reviewed.

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
