## [Reviewer comments · BMJ Open]

ARTICLE DETAILS

TITLE (PROVISIONAL)	Improving best practice for patients receiving hospital discharge letters: a realist review protocol
AUTHORS	Weetman, Katharine; Wong, Geoffrey; Scott, Emma; Schnurr, Stephanie; Dale, Jeremy

VERSION 1 – REVIEW

REVIEWER	Sonia Dalkin Northumbria University, UK
REVIEW RETURNED	05-Jul-2017

GENERAL COMMENTS	The article describes the protocol for a realist review of improving best practice for patients receiving hospital discharge letters. The protocol is clear, concise and methodologically sound. This is an interesting study, the findings of which will greatly contribute to health care practice. I look forward to further publications from this research. A few very minor notes are provided below but should not delay acceptance of the manuscript for publication: - Page 7, line 57, potential typo - repetition of 'and'.- Page 8, Further explanation as to why those with communicative needs are excluded is provided; a short explanation as to why the following is an exclusion criteria would potentially be beneficial to the reader: 'Discharge communications to units or physicians other than General Practitioners or family or community physicians e.g. another hospital'.
---

REVIEWER	Jean Macq UCL (Universite Catholique de Louvain) Belgium
REVIEW RETURNED	25-Jul-2017

GENERAL COMMENTS	First of all, thanks for giving me the opportunity to Review that protocol. Globally, it is nicely written and it is important to publish protocol of "realist Review" because of the "still new" methodological approach. I have one more global suggestion for change and 2 more detailed. 1) Line 6 to 9 of Page 9: it is said "As such, the stopping point for searching will be determined when 'theoretical saturation' is
--

	reached; when the addition of documents is not adding further knowledge (24)". I do not understand, how can we identify "the stopping point" before step 3 (document selection) and even step 4? There should be an iterative process whereby step2 search for a "limited" (something "manageable") number of article (starting from the most recent??? Or from review article?? Or???) that are then "screened" in step 3 and then "analysed for data extraction" in step 4. This process is repeated till "stopping point is reached". If this correspond to the process followed in the review, I would suggest to represent that "iterative cycle" (between steps 2, 3 and 4) in the figure making possible to identify the "stopping point". I think this is crucial in this type of review. Indeed, one may lose a lot of time in step 2 and 3, if the "cycle step 2-3-4" is not done. 2) Line 45 of page 7: the Step 1: Locate existing theories. It is unclear how the search for existing theories is done... before the search for articles. I understand that an initial program theory is build based on the prior knowledge of research team, but the difficulty with searching theories in literature is to clarify when the "unstructured" search stops (step 1) and when the structured search (step2) starts? 3) A detail... Line 34 of page 6: rather than "evaluation of an intervention" I would write "effectiveness evaluation of an intervention"
--	---

VERSION 1 – AUTHOR RESPONSE

Reviewer: 1

Reviewer Name: Sonia Dalkin

Institution and Country: Northumbria University, UK

Competing Interests: None declared

The article describes the protocol for a realist review of improving best practice for patients receiving hospital discharge letters. The protocol is clear, concise and methodologically sound. This is an interesting study, the findings of which will greatly contribute to health care practice. I look forward to further publications from this research.

A few very minor notes are provided below but should not delay acceptance of the manuscript for publication:

1. Page 7, line 57, potential typo - repetition of 'and'.

Response: This has been corrected.

2. Page 8 Further explanation as to why those with communicative needs are excluded is provided; a short explanation as to why the following is an exclusion criteria would potentially be beneficial to the reader: 'Discharge communications to units or physicians other than General Practitioners or family or community physicians e.g. another hospital'.

Response: We have added further information regarding this point and expanded on the reasoning for this exclusion:

“The first exclusion criterion states patient discharge communication to those other than GPs or family or community physicians will be excluded. This is because the review specifically focusses on discharge communication to GPs and patients rather than referrals or care-handovers. Furthermore, the review aims to develop a theory for patients receiving discharge communication and inclusion of hospital-hospital discharge may reduce clarity and produce a less focussed theory.”

Reviewer: 2

Reviewer Name: Jean Macq

Institution and Country: UCL (Universite Catholique de Louvain), Belgium

Competing Interests: none declared

First of all, thanks for giving me the opportunity to Review that protocol.

Globally, it is nicely written and it is important to publish protocol of "realist Review" because of the "still new" methodological approach.

I have one more global suggestion for change and 2 more detailed.

1) Line 6 to 9 of Page 9: it is said “As such, the stopping point for searching will be determined when ‘theoretical saturation’ is reached; when the addition of documents is not adding further knowledge (24)”.

I do not understand, how can we identify “the stopping point” before step 3 (document selection) and even step 4? There should be an iterative process whereby step2 search for a “limited” (something “manageable”) number of article (starting from the most recent??? Or from review article?? Or???) that are then “screened” in step 3 and then “analysed for data extraction” in step 4. This process is repeated till “stopping point is reached”.

If this correspond to the process followed in the review, I would suggest to represent that “iterative cycle” (between steps 2, 3 and 4) in the figure making possible to identify the “stopping point”. I think this is crucial in this type of review. Indeed, one may lose a lot of time in step 2 and 3, if the “cycle step 2-3-4” is not done.

Response: We agree that the cycle should include repetition of potentially more steps than just the searching. Consequently, we have removed the above from the searching section and instead added the following paragraph to step 6. This explains that any or all of the steps may be repeated in cycles in order to reach the “stopping point” i.e. theoretical saturation according to Pawson.

“After completion of step 6, any or all of the review steps may be revisited as necessary to refine the programme theory and attain “theoretical saturation”. The threshold for “theoretical saturation” will be decided according to Pawson’s ‘test of saturation’ (24, 35). Consequently, after each cycle of review steps, the research team will determine whether the latest cycle has provided additional information about the intervention to answer the research questions and test the programme theory (28). As such, the stopping point for the review will be determined when ‘theoretical saturation’ is reached; when the addition of documents and repetition of steps is not adding further knowledge (24).”

2) Line 45 of page 7: the Step 1: Locate existing theories. It is unclear how the search for existing theories is done... before the search for articles. I understand that an initial program theory is build based on the prior knowledge of research team, but the difficulty with searching theories in literature is to clarify when the “unstructured” search stops (step 1) and when the structured search (step2) starts?

Response: We have revised this paragraph and added in additional texts to clarify the process. Locating existing theories on how patients receiving written discharge communication is theorised to work or not work in different contexts will be completed through a scoping search by KW. The scoping search will be based on search terms centred on the intervention under study (e.g. patient copies/ receiving letters / discharge communication). This search will include a mix of electronic published resources (MEDLINE, Web of Science) and UK healthcare websites (Department of Health, Royal College of Physicians). Documents sourced within the scoping search will be interrogated for theories relating to patients receiving discharge communication. Theories located in the scoping stage will be inspected and selected based on relevance to the review aims and RQs; we seek theories which aid explanation of how and why patients receiving discharge communication results in different positive effects (e.g. drug adherence) and negative effects (e.g. preventable hospital readmissions). Any initial scoping search done to build the initial programme theory is not meant to be exhaustive but to function as a starting point for the realist review. During the review the initial programme theory is gradually developed. There are no hard and fast rules for how well developed the initial programme theory needs to be before the main searching is undertaken. Instead judgement is needed as is the need to balance the degree of comprehensiveness and practicalities. Our decision is that due to feasibility limitations, no more than 30 documents will be screened for theories in step 1. Any search strategies detailed in documents found from the scoping search will be used to inform step 2. "Keywords" "Medical Subject Headings (MeSH)" and any other indexing for the documents found will also be used to inform the step 2 searching phase. From the findings of the scoping search and utilising the various content expertise of the research team, an initial "rough" programme theory will be developed, to be refined throughout the realist review process. Once the initial programme theory has been developed, step 2, the structured formal searching phase, will commence.

3) A detail... Line 34 of page 6: rather than "evaluation of an intervention" I would write "effectiveness evaluation of an intervention"

Response: This has been changed.

VERSION 2 – REVIEW

REVIEWER	Jean Macq Universite Catholique de Louvain Belgium
REVIEW RETURNED	09-Oct-2017
GENERAL COMMENTS	I wish to congratulate authors for that nice protocol. I wish them the best luck in the implementation of the realist Review.